# Properties of INDETERMINATE DOMAIN Proteins from *Physcomitrium patens*: DNA-Binding, Interaction with GRAS Proteins, and Transcriptional Activity

**DOI:** 10.3390/genes14061249

**Published:** 2023-06-11

**Authors:** Saiful Islam Khan, Ren Yamada, Ryoichi Shiroma, Tatsuki Abe, Akiko Kozaki

**Affiliations:** 1Graduate School of Science and Technology, Shizuoka University, Ohya 836, Suruga-ku, Shizuoka 422-8021, Japan; 2Department of Biological Science, Faculty of Science, Shizuoka University, Ohya 836, Suruga-ku, Shizuoka 422-8021, Japan; 3Course of Bioscience, Department of Science, Graduate School of Integrated Science and Technology, Shizuoka University, Ohya 836, Suruga-ku, Shizuoka 422-8021, Japan

**Keywords:** INDETERMINATE DOMAIN (IDD) family, *Physcomitrium patens*, bryophytes, GRAS family, DELLA, SHORT-ROOT (SHR), gibberellic acid (GA), protein-protein interaction, transcription factor (TF)

## Abstract

INDETERMINATE DOMAIN (IDD) proteins are plant-specific transcription factors that interact with GRAS proteins, such as DELLA and SHORT ROOT (SHR), to regulate target genes. The combination of IDD and DELLA proteins regulates genes involved in gibberellic acid (GA) synthesis and GA signaling, whereas the combination of IDD with the complex of SHR and SCARECROW, another GRAS protein, regulates genes involved in root tissue formation. Previous bioinformatic research identified seven *IDDs*, two *DELLA*, and two *SHR* genes in *Physcomitrium patens*, a model organism for non-vascular plants (bryophytes), which lack a GA signaling pathway and roots. In this study, DNA-binding properties and protein–protein interaction of IDDs from *P. patens* (PpIDD) were analyzed. Our results showed that the DNA-binding properties of PpIDDs were largely conserved between moss and seed plants. Four PpIDDs showed interaction with *Arabidopsis* DELLA (AtDELLA) proteins but not with PpDELLAs, and one PpIDD showed interaction with PpSHR but not with AtSHR. Moreover, AtIDD10 (JACKDAW) interacted with PpSHR but not with PpDELLAs. Our results indicate that DELLA proteins have modified their structure to interact with IDD proteins during evolution from moss lineage to seed plants, whereas the interaction of IDD and SHR was already present in moss lineage.

## 1. Introduction

*Physcomitrium patens* is a model organism for non-vascular plants (bryophytes) and is used to study the evolution of land plants [1,2]. Although the morphology and life cycle of mosses are different from those of vascular plants, some signal transduction systems and the network of transcription factors in vascular plats are conserved in *P. patens* [2]. For example, some hormones and their signal transductions, such as auxin, cytokinin, abscisic acid, and ethylene, are conserved in *P. patens* [3,4,5,6]. However, the gibberellic acid (GA) signaling pathway is not conserved in *P. patens* [7,8], although it contains an ent-kaurene synthase, an enzyme involved in GA biosynthesis in seed plants [9].

*INDETERMINATE DOMAIN* (*IDD*) genes that encode transcription factors containing conserved IDDs with four zinc fingers (ZFs) comprise a conserved gene family across plants [10,11]. Recent studies have revealed that IDD transcription factors support various biological functions [12,13], synthesis and signaling of GA [14,15] and auxin [16], root development [17,18], seed development [19], the immune system [20], and so on [13]. Some IDD transcription factors regulate target genes by interacting with GRAS family transcription factors, such as DELLA proteins and SHORT ROOT (SHR) [14,15,17,21].

In *Arabidopsis*, AtIDD2 (GAF1), AtIDD3 (MGP), and other AtIDDs interact with AtDELLAs, such as GIBBERELLIC ACID INSENSITIVE (GAI) or REPRESSOR of *ga1-3* (RGA), to regulate genes involved in GA synthesis (*GA3ox1* and *GA20ox1*) and GA signaling (*SCL3*) [14,15,21]. In rice plants, a complex of SLENDER RICE 1 (SLR1: rice DELLA) and OsIDD2 regulates the expression of miR396, whose target is growth-regulating factors (OsGRFs) [22]. The GRF family protein is a plant-specific transcription factor involved in many aspects of plant development and growth, including GA signaling and stem elongation [23,24,25,26]. These results indicate that the complex of IDD and DELLA proteins is primarily involved in the regulation of GA synthesis and signaling in angiosperms.

During GA signaling, DELLAs interact with the GA receptor GA-INSENSITIVE DWARF1 (GID1) when active GA binds to GID1. The GID1-DELLA complex is interacted by a particular F-box protein (SLY1 in *Arabidopsis* or GID2 in rice), and the GA-GID1-DELLA-F-box complex is recruited by the SCF ubiquitin E3 ligase complex and then degraded by the 26S proteasome in angiosperms [27,28,29,30].

Bioinformatic analysis identified two genes for DELLA and two genes for GID1-related proteins (GLPs) in *P. patens* [7,8]. Although PpDELLAs showed a highly conserved GRAS domain, they lack the two highly conserved domains at the N-termini among angiosperm DELLAs [7,8], which are necessary for GID1-DELLA interaction [29,31,32]. The absence of interaction between PpDELLAs and PpGLPs or AtGID1 indicated that GID1-DELLA-mediated GA signaling is not present in *P. patens* [7,8]. A recent study has shown that PpDELLA proteins are involved in the regulation of spore germination and reproductive development [33].

AtIDD10 (JKD) and MGP interact with the SCR-SHR complex and regulate the target genes, including *AtSCR* [17,34,35]. AtSCR and AtSHR are GRAS family transcription factors, which are involved in the radial pattern formation of ground tissues in *Arabidopsis* [36,37,38,39]. In silico analysis identified two genes for SHR and three genes for SCR in *P. patens* [40]. Recent research has shown that the interaction between PpSHR and PpSCR is conserved, and PpSHR and PpSCR are involved in the regulation of cell division patterning together with another GRAS protein, namely, the LATERAL SUPPRESSOR ortholog [40].

Recent genomic search identified seven *IDD* genes in *P. patens* [11,13]. However, the function of *IDD* genes in *P. patens* has not been investigated. Considering that IDD proteins function by interacting with DELLA and SHR to play an important role in vascular plant-specific development and signaling systems, development of the root and vascular system, GA synthesis, and signaling [13,14,15,17,21,34,35], IDD proteins might play an important role in the evolution of land plants. In elucidating the mechanism by which IDD proteins play a role in the evolution of land plants, the function of PpIDDs must be comprehensively understood. In revealing the function of PpIDDs, the biochemical properties of PpIDDs, DNA-binding properties, interaction with PpDELLA and PpSHR, and the effect of interaction with GRAS proteins on the transcriptional activity of PpIDDs were investigated.

## 2. Materials and Methods

### 2.1. Plant Materials and Growth Conditions

*P. patens* (Gransden2004 strain) [41,42] was provided by Dr. Hasebe (National Institute for Basic Biology, Okazaki, Japan). The moss was grown under sterile conditions on BCDAT medium at 25 °C under continuous light [43].

### 2.2. Cloning of PpIDD and PpGRAS Genes

The sequence of PpIDDs and PpGRAS was obtained from the Phytozome 13 database (Phytozome (doe.gov), 1 August 2022). The accession numbers of PpIDDs were listed as follows: PpIDD1 (Pp3c1-16920), PpIDD2 (Pp3c1-20230), PpIDD3 (Pp3c2-20240), PpIDD4 (Pp3c2-22070), PpIDD5 (Pp3c5-20980), PpIDD6 (Pp3c6-10870), and PpIDD7 (Pp3c6-10890).

The sequence alignment in Appendix A was made using MEGA-11 and ClustalW.

Total RNA was purified from gametophores of *P. patens* with ISOGEN (NIPPON GENE, Toyama, Japan). cDNA was synthesized using ReverTra Ace (TOYOBO, Osaka, Japan). The coding sequence (CDS) of each gene was amplified by PCR using PrimeSTAR HD DNA polymerase (Takara Bio Inc., Shiga, Japan) and synthesized cDNA as a temperate. The primer sequences used for PCR are described in Appendix A. Amplified PCR products were introduced in PCR-Blunt (Thermo Fisher, Pittsburgh, PA, USA), and the sequence was confirmed. Cloned cDNAs were used for vector construction.

### 2.3. Yeast One-Hybrid Assay

The yeast one-hybrid assay was performed using the Matchmaker One-Hybrid system (Clontech, Mountain View, CA, USA). The reporter genes used in this experiment were described in a previous study [34]. Four tandem repeat copies of IDDBS (i.e., the IDD-binding sequence, TTTGTCGTATT) [44] were cloned into the cloning site of the pLacZi vector (Clontech). The reporter gene was transferred to the genomic DNA of the yeast strain BY5444.

Prey vectors were constructed by introducing the CDS of each gene into the cloning site of pGAD424 plasmids (Clontech). The prey vectors were introduced into yeast carrying IDDBS. The X-gal activities were detected by a filter assay in accordance with the manufacturer’s manual (Clontech).

### 2.4. Yeast Two-Hybrid (Y2H) Assay

pGAD424 and pGBT9 plasmids (Clontech) were used to construct prey and bait vectors, respectively. Prey and bait vectors were transformed into the yeast strain PJ69-4A.

Yeast cells were cultured in liquid media without Leu and Trp (−LW) until reaching a density of 0.4–0.6 OD/mL. Cells were diluted to 6 × 10^2^ cells/μL, followed by further 10-fold serial dilutions. Diluted cells (10 μL) were spotted on −LW and −LWH (i.e., media without Leu, Trp, and His). As needed, appropriate concentration of 3-amino-1,2,4-triazole was added to the plates. Cells were grown for 3 days at 30 °C.

### 2.5. Transient Assay

The effector vector for the transient assay was constructed using the pUC19 vector (35S:pUC19). The CDS of each gene was inserted downstream of the CaMV35S promoter of 35S:pUC19 [34].

In constructing effector vectors with a DNA-binding domain of a yeast GAL4 transcription factor (GAL4BD), sequences for fusion proteins (JKD, PpIDD2, 4, and 5 with GAL4BD) were amplified by PCR using pGBT9 carrying each gene as a temperate and were inserted downstream of the CaMV35S promoter of 35S:pUC19.

Transient assays were carried out as previously described [34]. Protoplasts (150 μL; 10^7^ protoplasts mL^−1^) isolated from *Arabidopsis* T87 cell culture were co-transfected with 10 μg each of the effector and reporter plasmid DNA and 5 μg of the internal control plasmid (35S:hRLUC). After incubating the protoplasts at 22 °C for 20 h, they were collected and used to measure the activities. LUC and hRLUC activities were determined using the Dual-Luciferase Reporter Assay system (Promega, Madison, WI, USA). The LUC activity was normalized according to the hRLUC activity. By comparing this ratio with that obtained using an empty vector, the relative ratio was determined. From four independent experiments, the mean of relative ratios was calculated and statistically analyzed using the Student’s *t*-test or one-way ANOVA followed by Tukey’s test (*p* < 0.05).

## 3. Results

### 3.1. IDD Gene Family in P. patens

Bioinformatic analysis identified seven PpIDDs, including PpIDD1 (Pp3c1-16920), PpIDD2 (Pp3c1-20230), PpIDD3 (Pp3c2-20240), PpIDD4 (Pp3c2-22070), PpIDD5 (Pp3c5-20980), PpIDD6 (Pp3c6-10870), and PpIDD7 (Pp3c6-10890) [11,13]. The number of PpIDDs is less than that of seed plants (*Arabidopsis*: 16, *Oryza sativa*: 15, etc.). The amino acid length of most PpIDDs (PpIDD1: 770 aa; PpIDD2: 777 aa; PpIDD3: 763 aa; PpIDD4: 809 aa; PpIDD5: 868 aa; PpIDD6 and 7: 474) was longer than that of *Arabidopsis* (from 385 to 602 aa) and *O. sativa* (from 459 to 633 aa). Phylogenetic analysis showed that PpIDD5, 6, and 7 belonged to the group of AtIDD14, 15, and 16 (A-group IDD based on the classification by Colasanti et al. (2006) [10]) and other four PpIDDs were in the same clade as MGP [11,13].

The nucleotide sequence of *PpIDD6* and *PpIDD7* genes was completely identical, which indicated that these genes were duplicated. Gene structure analysis showed that *PpIDD1–4* contains three exons and the region corresponding to the fourth ZF was divided by an intron, which is similar to those of most *IDD* genes from *Arabidopsis*, rice, and corn [10]; *PpIDD5* contains six exons, and *PpIDD6* and *7* contain four exons and the first ZF of these genes was on the separated exon from that containing other ZFs (Figure 1).

The amino acid sequence of the IDD domain of PpIDD1–4 was well conserved, although that of PpIDD5, 6, and 7 was slightly different from other IDDs and closer to that of A-group IDDs (Appendix A). Moreover, mutations were observed in the third and fourth ZFs in the ID domain of PpIDD6 and 7 (Appendix A).

Most of the IDD proteins except for A-group IDDs contain two conserved motifs at C-terminal regions, namely, MSATALLQKAA and T[R/L]DFLG [10,11]. Both motifs were conserved in PpIDD1–4 (Appendix A). In addition, PpIDD1–4 contains 30–60 aa of a Q-rich region between conserved MSATALLQKAA and T[R/L]DFLG (Appendix A).

cDNAs were isolated for PpIDD 1–5, whereas PpIDD6 and 7 cDNA could not be isolated. Therefore, PpIDD1–5 was used hereafter.

### 3.2. DNA-Binding Properties of PpIDD Proteins

All AtIDDs and some IDD proteins from other plants could bind to the IDD-binding sequence 5′-TTTGTC(G/C)(T/C)(T/a)(T/a)T-3′ (IDDBS) [13,14,21,45,46].

Whether each PpIDD protein binds to IDDBS was examined by using a yeast one-hybrid assay. PpIDD1–4 bound to IDDBS, whereas PpIDD5 did not show binding to IDDBS (Figure 2).

### 3.3. Analysis of Protein–Protein Interaction of PpIDD Proteins

IDD proteins from *Arabidopsis*, rice, and other plants could interact with GRAS proteins, such as DELLA and SHR [14,15,17,21,22,47,48]. We examined whether PpIDD proteins interact with GRAS proteins from *P. patens* and *Arabidopsis* by using a Y2H assay. In *P. patens*, two *DELLA* genes and two *SHR* genes were identified [7,8,40,49], and cDNA for PpDELLAa and PpDELLAb and one for PpSHR (PpSHR2) were obtained. Therefore, these cDNAs were used to construct vectors for the Y2H assay. We used RGA and GAI as *Arabidopsis* DELLA proteins and AtSHR in this experiment.

PpIDD5 showed relatively strong interaction and PpIDD2 and 4 showed weak interaction with RGA and GAI but not with PpDELLAs (Figure 3B,D,E). Because the background activity of PpIDD1 and PpIDD3 was very high, it was difficult to detect the interaction between these proteins with GRAS proteins (Figure 3A,C). However, a very faint interaction was detected between PpIDD3 and GAI or RGA (Figure 3C). It was unexpected that PpIDD5 interacted with RGA and GAI because A-group IDDs in *Arabidopsis* did not interact with any AtDELLA proteins [21].

Moreover, PpIDD2 showed a weak interaction with PpSHR2 but not with AtSHR (Figure 3B). We also examined whether JKD interacts with PpDELLAs and PpSHR2. Interestingly, JKD interacted with PpSHR2 but not with PpDELLA (Figure 3F,G).

### 3.4. Transcriptional Activities of PpIDD

Next, we examined the transcriptional activity of PpIDDs and the effect of the addition of DELLA or SHR proteins on the activity by conducting transient assay using *Arabidopsis* culture cells. In the Y2H assay, PpIDD4 and PpIDD5 interacted with AtDELLAs, whereas PpIDD2 interacted with both AtDELLAs and PpSHR2 (Figure 3). Therefore, these PpIDDs were used for the transient assay. In addition, because JKD interacted with PpSHR2, JKD was used in this experiment. The target promoter of these PpIDDs has not been identified; thus, PpIDDs fused with the GAL4 DNA-binding domain (GAL4BD) were used as effectors, and a luciferase reporter gene with GAL4-responsive elements was used as the reporter gene (pGLL) (Figure 4A). As the positive control, JKD fused with GAL4BD (GBD JKD) in combination with RGA was used in this experiment.

We could not detect any activity when GBD PpIDD2, GBD PpIDD4, and GBD PpIDD5 were used as effectors, although GBD JKD showed moderate-level activity (Figure 4B). The addition of RGA did not affect the activity of PpIDD2, 4, or 5, although the activity of GBD JKD was increased by the addition of RGA (Figure 4B). The addition of PpSHR2 slightly increased the activity of GBD PpIDD2, although there was no significant difference. The activity of BGD JKD was not affected by the addition of PpSHR2 (Figure 4B), whereas the combination of GBD JKD and AtSHR showed approximately three times the transcriptional activity of GBD JKD alone in a previous study [34].

## 4. Discussion

In *P. patens*, seven genes for IDD proteins were identified by performing genomic research [11,13]. Two of these proteins, namely, PpIDD6 and PpIDD7, were encoded with an IDD domain in which the third and fourth ZFs are mutated (Appendix A). However, we could not isolate these genes and examine the properties of these proteins. Considering that the second and third ZFs are required for DNA-binding [44,50], we hypothesized that PpIDD6 and PpIDD7 cannot bind to IDDBS. The exon–intron structure of the *PpIDD1–4* was largely conserved among *IDD* genes from other plants, whereas that of *PpIDD5*–*7* was slightly different from the others; the first ZF was on an exon separated from the others (Figure 1). Although *PpIDD5–7* were classified to A group *IDD*, the exon-intron structure of A group IDD from *Arabidopsis*, rice, and maize was slightly different; the first ZF was on the same exon as other ZFs [10].

DNA-binding analysis of PpIDDs by Y1H assay showed that PpIDD 1–4 bound to 11 bp of IDDBS, whereas PpIDD 5 did not bind to IDDBS (Figure 2). A previous study showed that the binding affinity of AtIDD14, 15, and 16 to IDDBS was slightly weaker than that of other AtIDDs [21]. Considering that the amino acid sequence of the ID domain of A-group IDDs is slightly different from other IDDs, it may cause a difference in DNA-binding affinity to IDDBS.

Our Y2H results showing that PpIDD2, 4, and 5 interacted with RGA and GAI but not with PpDELLAs (Figure 3) indicated that DELLA has modified its structure to interact with IDDs during evolution.

With regard to the interaction with DELLA proteins, the conserved sequences in the C-terminal region, namely, the MSATALLQKAA motif and T[R/L]DFLG motif, are considered important [14,15]. These sequences are well conserved among IDD proteins in *Arabidopsis*, rice, and maize and so on, except for A-group IDDs [10,11,13]. Although these sequences are also conserved in PpIDDs except for PpIDD 5–7, PpIDD1 did not interact with AtDELLAs. However, PpIDD5 showed a relatively strong interaction with AtDELLAs, even though the amino acid sequence of the C-terminal region is quite different from other IDD proteins. Moreover, A-group AtIDDs (AtIDD14, 15, and 16) did not interact with AtDELLAs [21]. The result indicated that a merely conserved amino acid sequence in IDD proteins cannot establish the interaction of IDD proteins with DELLA proteins.

On the contrary, the SAW domain in the C-terminal region of GAI is required for interaction with AtIDD2 [15]. With regard to the interaction of RGA with MGP, the leucine-rich region I (LHR1) domain is essential but not sufficient [14]. Although PpDELLAs lack the DELLA domain at the N-terminal region, the C-terminal region, including LHR1 and SAW, is well conserved [7,8]. However, PpDELLA did not interact with either PpIDDs or JKD. Further investigations must be conducted to identify the sequence essential for the interaction of IDDs and DELLAs and to elucidate the mechanism by which DELLA proteins have changed their structure to interact with IDD proteins.

In contrast to PpDELLA, PpSHR2 interacted with PpIDD2 and JKD (Figure 3). With regard to the interaction with SHR, ZF3 and ZF4 in the ID domain are considered important, and the SHR-binding motif (R(R/K)DxxITHxAFCD) is necessary [50,51]. The IDD domain, including a SHR-binding motif, is well conserved among PpIDDs except for the A-group PpIDD5–7. However, PpIDD1, 3, and 4 did not interact with either PpSHR2 or AtSHR. Moreover, PpIDD2 interacted with PpSHR2 but not with AtSHR. In addition, JKD showed interaction with PpSHR2 with slightly weaker affinity than that with AtSHR. The results indicated that the interaction of IDD and SHR was already present in the moss lineage, and IDDs and SHR have evolved their structure to interact tightly with each other.

In our previous study, the transcriptional activity of AtIDDs was enhanced by the addition of AtDELLA or AtSHR [34,35]. Although some PpIDDs interacted with AtDELLAs, we could not observe either their transcriptional activity or the enhancement effect on the transcriptional activity of PpIDDs by interacting with DELLA proteins in transient assays (Figure 4). PpSHR2 did not enhance the activity of JKD and PpIDD2, although the activity of PpIDD2 was slightly increased by the addition of PpSHR2 (Figure 4). Although we could not find a statistical difference between the activity of PpIDD2 alone and the combination PpIDD2 and PpSHR2 in this study, it is possible to detect the enhancement effect of PpSHR2 on the activity of PpIDD2 with an improvement in experimental methods.

The transcriptional activity of the combination of PpIDDs and GRAS proteins was not detected in the transient assay because of several possible reasons. GRAS proteins naturally do not enhance the transcriptional activity of PpIDDs by interacting with one another; GRAS proteins serve as repressors rather than enhancers by interacting with PpIDDs. The interaction of these proteins was too weak to affect the activity; PpIDDs did not function properly because the transient assay was conducted by using *Arabidopsis* cells. Further investigation must be conducted to clarify the mechanism by which DELLAs and SHR have become the co-activators of IDD proteins.

Although PpDELLAs did not interact with either PpGLP or GID1 from *Arabidopsis* and rice, DELLA and GID from lycophyte (*Selaginella kraussiana* and *Selaginella moellendorffii*) interacted in a GA-dependent manner [7,8]. The results indicated that the GA-stimulated interaction of GID1 and DELLA appeared in the land-plant lineage after the bryophyte divergence (~430 million years ago) [52]. The lycophyte is an old lineage of vascular plants containing active GA and GA signaling components [7,8,53]. Therefore, the interaction of IDD and GRAS proteins in lycophytes must be examined to clarify the relationship between the interaction of IDDs with GRAS proteins, GA signaling, and root development. Finally, the investigation of the biological function of PpIDDs will provide useful information to understand the evolution of land plants.

## 5. Conclusions

We isolated five *IDD* genes from *P. patens*; the amino acid sequence of the IDD domain of PpIDD1–4 was well conserved, whereas that of PpIDD5 was slightly different and close to that of A-group IDDs in angiosperm, including AtIDD14–16 and OsIDD12–14. PpIDD1–4 but not PpIDD5 showed binding to 11 bp of the IDD binding sequence. Analysis of protein-protein interaction using the Y2H system showed that PpIDD2, 4, and 5 interacted with AtDELLAs, RGA, and GAI but not with PpDELLAs, and PpIDD2 interacted with PpSHR2 but not with AtSHR. On the other hand, AtIDD10 (JACKDAW) interacted with both AtSHR and PpSHR. These results indicate that DELLA proteins have modified their structure to interact with IDD proteins during evolution from the moss lineage to seed plants, whereas SHR had potential to interact with IDD proteins from the moss lineage. Although some PpIDDs showed interaction with RGA or PpSHR, the transcriptional activity of PpIDDs was not increased by the addition of RGA or PpSHR in a transient assay using *Arabidopsis* cells. Further research is required to clarify the function of the interaction between these proteins. This research provides useful information about the evolution of the complex of IDD and GRAS proteins, which play important biological roles in vascular plants such as GA synthesis, GA signaling, and root development.

## Figures and Tables

**Figure 1 genes-14-01249-f001:**
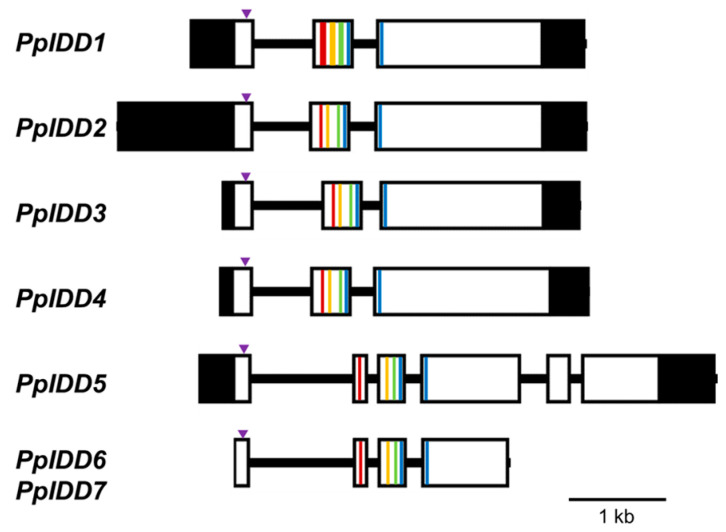
Schematic diagram of the structure of *IDD* genes from *P. patens.* The boxes indicate exons and black lines between boxes indicate introns. The 5′- and 3′-untranslated regions are indicated by the dark shaded boxes. The putative NLS is indicated by inverted triangles. Each zinc finger is shown by color bars (ZF1, red; ZF2, orange; ZF3, green; and ZF4, blue). The ZF4 is divided by an intron in all *PpIDD* genes.

**Figure 2 genes-14-01249-f002:**
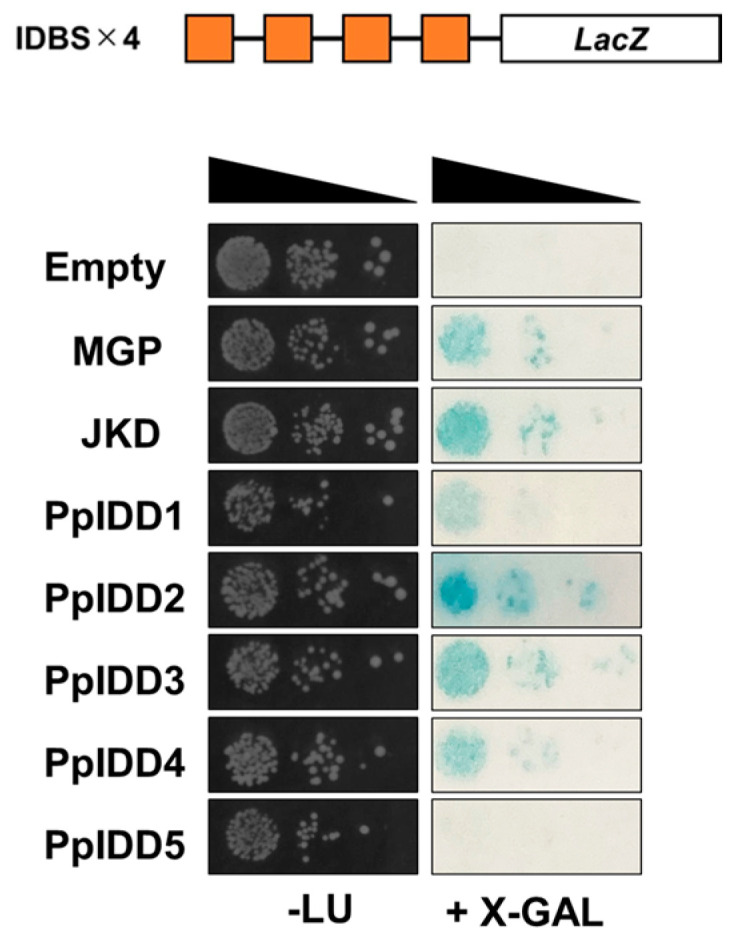
DNA binding ability of PpIDDs was analyzed using yeast one-hybrid assay. Four tandem-repeats of IDD binding sequence (IDBS) were inserted upstream of the *LacZ* gene and used as promoters. DNA binding was detected by X-gal staining assay. Empty, empty vector pGAD424.

**Figure 3 genes-14-01249-f003:**
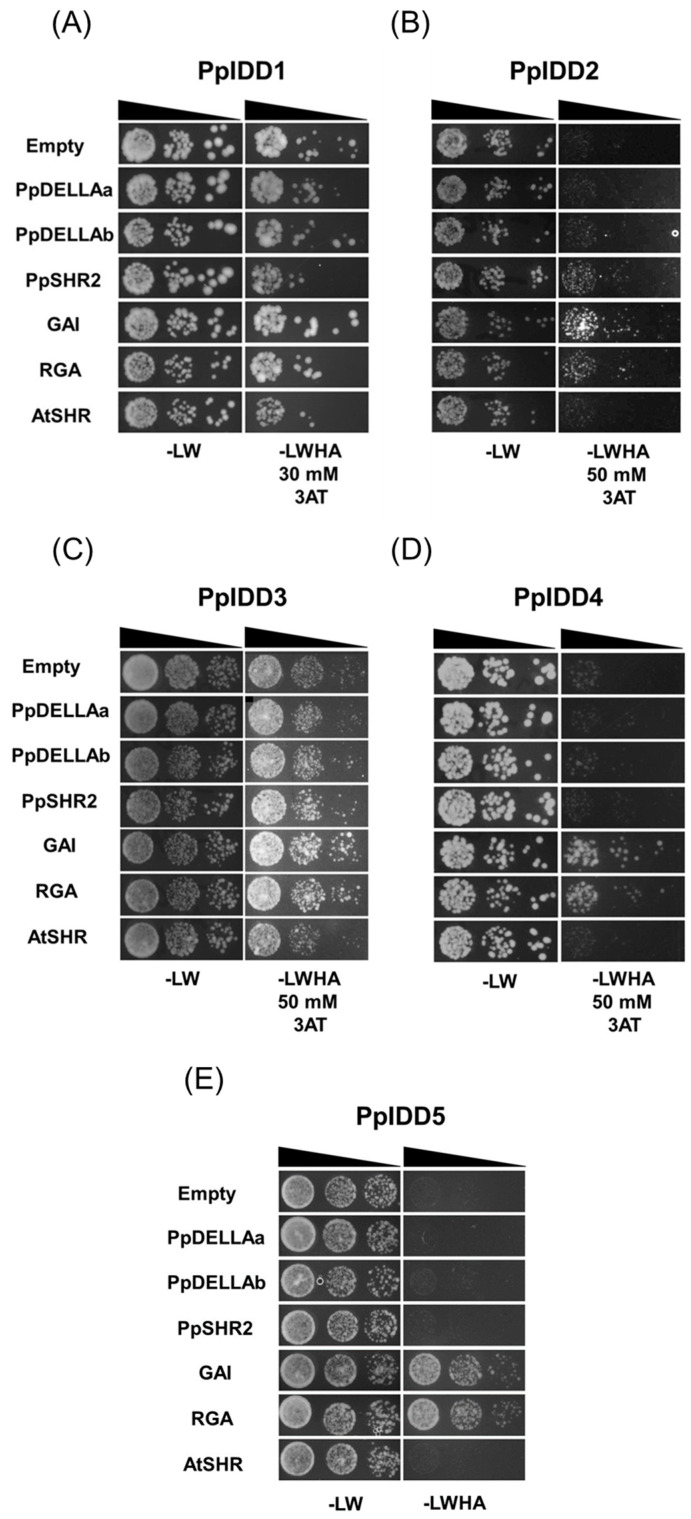
Protein-protein interaction of PpIDD proteins and JKD with GRAS proteins was analyzed using yeast two-hybrid assay. The coding sequences (CDSs) for (**A**) PpIDD1, (**B**) PpIDD2, (**C**) PpIDD3, (**D**) PpIDD4, (**E**) PpIDD5, and (**F**,**G**) JKD in pGBT9 were used as baits. The CDSs for PpDELLAa, PpDELLAb, PpSHR2, GAI, RGA, and AtSHR in pGAD424 were used as prey. Both prey and bait vectors were transformed into yeast cells that were grown on −LW or −LWH with an appropriate concentration of 3-amino-1,2,4 triazole (3-AT). Empty, empty vector pGAD424.

**Figure 4 genes-14-01249-f004:**
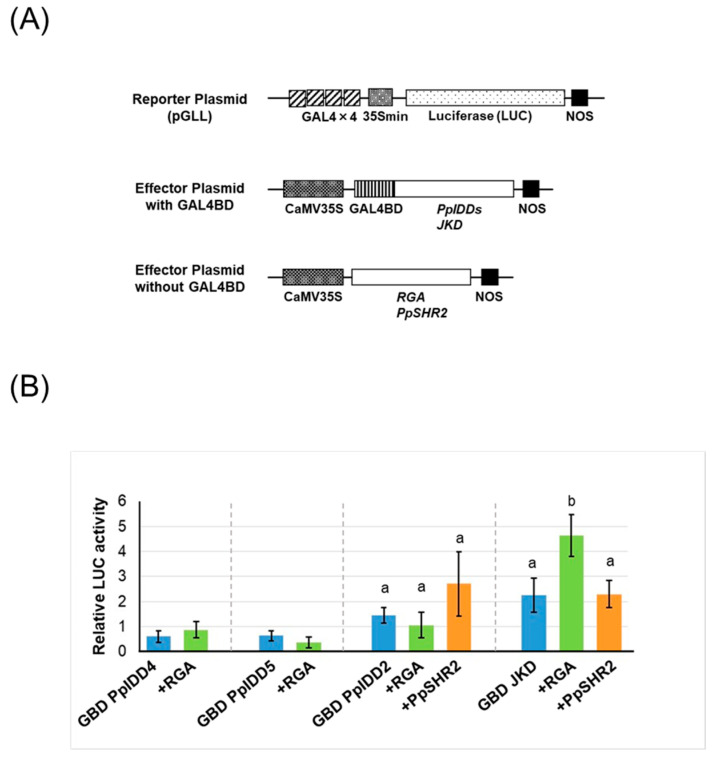
Transcriptional activity of the combination of PpIDD proteins and GRAS proteins. (**A**) Sketch map of reporter and effector plasmids used in transient assays. pGLL is a GAL4-responsive reporter vector containing four copies of the GAL4 binding site, minimal CaMV 35S promoter (35Smin), the gene for luciferase (LUC), and nopaline synthase (NOS) terminator. Effector plasmid with GAL4 DNA binding domain (GAL4BD) contains genes for fusion protein of GAL4BD with either PpIDDs or JKD, and effector plasmid without GAL4BD contains either *RGA* or *PpSHR2* genes. The genes in both effector vectors were driven by CaMV35S promoter. (**B**) Transcription activity of PpIDDs with GAL4BD (GBD PpIDDs) and GBD JKD in combination with GRAS proteins. An empty vector was used as a control, and all LUC activities were expressed relative to this control (value set at 1). Values shown are the average of results from four independent experiments. Error bars represent standard deviation (SD). Statistical analysis was performed separately for each IDD protein; data with GBD PpIDD4 and GBD PpIDD5 were analyzed using Student’s *t*-test, and data with GBD PpIDD2 and GBD JKD were analyzed using one-way ANOVA followed by Tukey’s test (*p* < 0.05).

## Data Availability

Not applicable.

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
