# Peer review of "Properties of INDETERMINATE DOMAIN Proteins from Physcomitrium patens: DNA-Binding, Interaction with GRAS Proteins, and Transcriptional Activity"

_genes, 2023, doi:10.3390/genes14061249_

Round 1
Reviewer 1 Report
The manuscript describes the study of INDETERMINATE DOMAIN (IDD) in in Physcomitrium patens which was bioinformatic analyzed. The authors addressed the DNA-binding properties and the protein-protein interaction activities using Y1H and Y2H respectively. This is a rather short article, however the authors were able to verify the DNA binding ability and protein-protein interaction for some of the PpIDD proteins. In the later part of the manuscript, the authors also demonstrated the transcriptional activity of PpIDD proteins, although there was no significant conclusion in this part.
The experimental design of the manuscript is reasonable and it does provide experimental proof of the DNA/protein interaction of PpIDDs.
Minor comments:
The author may want to provide a short conclusion for the manuscript.
Author Response
2023/06/02
Response to Reviewers:
We are most grateful to you and the reviewers for the helpful comments on the original version of our manuscript. We have addressed all the comments made by the reviewers. We hope that the explanations and revisions of our work are satisfactory.
Reviewer1
- The author may want to provide a short conclusion for the manuscript.
- As the reviewer suggested, we added a conclusion section.
Reviewer 2 Report
In this study, the authors have identified IDDs, DELLA and SHR genes in model organism Physcomitrium patens. And they further to analyze DNA-binding properties and protein–protein interaction of IDDs from P. patens. This work provides essential information of IDDs in moss lineage, which will help us to understand the regulation of IDDs in plants. However, several points still need to be addressed before accepted. My decision is "Major Revision".
1. All the figures in the manuscript are of low resolution.
2. Suggest to conduct phylogenetic analysis of IDDs, DELLA and SHR genes in moss lineages and plants.
3. How about the expression model of IDDs in P. patens.
Author Response
2023/06/02
Response to Reviewers:
We are most grateful to you and the reviewers for the helpful comments on the original version of our manuscript. We have addressed all the comments made by the reviewers. We hope that the explanations and revisions of our work are satisfactory.
Reviewer2
- All the figures in the manuscript are of low resolution.
- As the reviewer requested, we replaced the figures by the figures with higher resolution.
- Suggest to conduct phylogenetic analysis of IDDs, DELLA and SHR genes in moss lineages and plants.
- As the reviewer suggested, we also though that we have to conduct phylogenetic analysis of IDDs, DELLA, and SHR. However, previous research that we cited ([11] Prochetto, S. et al. annals of Botany 2020,126,85-101; [13] Kumar, M. et al. Int J Mol Sci 2019, 20; [7] Hirano, K. et al. The Plant Cell 2007, 19, 3058-3079; Yasumura,Y. et al. Current Biology 2007, 17, 1225-1230; [40] Ishikawa, M. et al. Proceedings of the National Academy of Sciences 2023, 120,; [49] Phokas, A and Coates, J.C., Evolution & Development 2021, 23, 137-154 ) performed the detailed phylogenetic analysis and we thought that we cannot carry out better analysis . Therefore, we concentrated on biochemical analysis in this research.
- How about the expression model of IDDs in P. patens.
- As the reviewer mentioned, expression analysis of IDD genes in P. patens is very important. Although we could not analyze the expression of IDD genes in P. patens in this research, we will carry out tissue level analysis of IDD expression in P. patens in future research.
Round 2
Reviewer 2 Report
NA